# Saliency-based Sequential Image Attention with Multiset Prediction

**Sean Welleck**
New York University
wellecks@nyu.edu

**Jialin Mao**
New York University
jialin.mao@nyu.edu

**Kyunghyun Cho**
New York University
kyunghyun.cho@nyu.edu

**Zheng Zhang**
New York University
zz@nyu.edu

## Abstract

Humans process visual scenes selectively and sequentially using attention. Central to models of human visual attention is the saliency map. We propose a hierarchical visual architecture that operates on a saliency map and uses a novel attention mechanism to sequentially focus on salient regions and take additional glimpses within those regions. The architecture is motivated by human visual attention, and is used for multi-label image classification on a novel multiset task, demonstrating that it achieves high precision and recall while localizing objects with its attention. Unlike conventional multi-label image classification models, the model supports *multiset* prediction due to a reinforcement-learning based training process that allows for arbitrary label permutation and multiple instances per label.

## 1 Introduction

Humans can rapidly process complex scenes containing multiple objects despite having limited computational resources. The visual system uses various forms of attention to prioritize and selectively process subsets of the vast amount of visual input [6]. Computational models and various forms of psychophysical and neuro-biological evidence suggest that this process may be implemented using various "maps" that topographically encode the relevance of locations in the visual field [17, 39, 13].

Under these models, visual input is compiled into a *saliency-map* that encodes the conspicuity of locations based on bottom-up features, computed in a parallel, feed-forward process [20, 17]. Top-down, goal-specific relevance of locations is then incorporated to form a *priority map*, which is then used to select the next target of attention [39]. Thus processing a scene with multiple attentional shifts may be interpreted as a feed-forward process followed by sequential, recurrent stages [23]. Furthermore, the allocation of attention can be separated into *covert* attention, which is deployed to regions without eye movement and precedes eye movements, and *overt* attention associated with an eye movement [6]. Despite their evident importance to human visual attention, the notions of incorporating saliency to decide attentional targets, integrating covert and overt attention mechanisms, and using multiple, sequential shifts while processing a scene have not been fully addressed by modern deep learning architectures.

Motivated by the model of Itti et al. [17], we propose a hierarchical visual architecture that operates on a saliency map computed by a feed-forward process, followed by a recurrent process that uses a combination of covert and overt attention mechanisms to sequentially focus on relevant regions and take additional glimpses within those regions. We propose a novel attention mechanism for implementing the covert attention. Here, the architecture is used for multi-label image classifica-

tion. Unlike conventional multi-label image classification models, this model can perform *multiset* classification due to the proposed reinforcement-learning based training.

## 2 Related Work

We first introduce relevant concepts from biological visual attention, then contextualize work in deep learning related to visual attention, saliency, and hierarchical reinforcement learning (RL). We observe that current deep learning models either exclusively focus on bottom-up, feed-forward attention or overt sequential attention, and that saliency has traditionally been studied separately from object recognition.

### 2.1 Biological Visual Attention

Visual attention can be classified into *covert* and *overt* components. Covert attention precedes eye movements, and is intuitively used to monitor the environment and guide eye movements to salient regions [6, 21]. Two particular functions of covert attention motivate the Gaussian attention mechanism proposed below: noise exclusion, which modifies perceptual filters to enhance the signal portion of the stimulus and mitigate the noise; and distractor suppression, which refers to suppressing the representation strength outside an attention area [6]. Further inspiring the proposed attention mechanism is evidence from cueing [1], multiple object tracking [8], and fMRI [30] studies, which indicate that covert attention can be deployed to *multiple*, *disjoint* regions that *vary in size* and can be conceptually viewed as multiple "spotlights".

Overt attention is associated with an eye movement, so that the attentional focus coincides with the fovea's line of sight. The planning of eye movements is thought to be influenced by bottom-up (scene dependent) saliency as well as top-down (goal relevant) factors [21]. In particular, one major view is that two types of maps, the saliency map and the priority map, encode measures used to determine the target of attention [39]. Under this view, visual input is processed into a feature-agnostic saliency map that quantifies distinctiveness of a location relative to other locations in the scene based on bottom-up properties. The saliency map is then integrated to include top-down information, resulting in a priority map.

The saliency map was initially proposed by Koch & Ullman [20], then implemented in a computational model by Itti [17]. In their model, saliency is determined by relative feature differences and compiled into a "master saliency map". Attentional selection then consists of directing a fixed-sized attentional region to the area of highest saliency, i.e. in a "winner-take-all" process. The attended location's saliency is then suppressed, and the process repeats, so that multiple attentional shifts can occur following a single feed-forward computation.

Subsequent research effort has been directed at finding neural correlates of the saliency map and priority map. Some proposed areas for salience computation include the superficial layers of the superior colliculus (sSC) and inferior sections of the pulvinar (PI), and for priority map computation include the frontal eye field (FEF) and deeper layers of the superior colliculus (dSC)[39]. Here, we need to only assume existence of the maps as conceptual mechanisms involved in influencing visual attention and refer the reader to [39] for a recent review.

We explore two aspects of Itti's model within the context of modern deep learning-based vision: the use of a bottom-up, featureless saliency map to guide attention, and the sequential shifting of attention to multiple regions. Furthermore, our model incorporates top-down signals with the bottom-up saliency map to create a priority map, and includes covert and overt attention mechanisms.

### 2.2 Visual Attention, Saliency, and Hierarchical RL in Deep Learning

Visual attention is a major area of interest in deep learning; existing work can be separated into sequential attention and bottom-up feed-forward attention. Sequential attention models choose a series of attention regions. Larochelle & Hinton [24] used a RBM to classify images with a sequence of fovea-like glimpses, while the Recurrent Attention Model (RAM) of Mnih et al. [31] posed single-object image classification as a reinforcement learning problem, where a policy chooses the sequence of glimpses that maximizes classification accuracy. This "hard attention" mechanism developed in [31] has since been widely used [27, 44, 35, 2]. Notably, an extension to multiple

objects was made in the DRAM model [3], but DRAM is limited to datasets with a natural label ordering, such as SVHN [32]. Recently, Cheung et al. [9] developed a variable-sized glimpse inspired by biological vision, incorporating it into a simple RNN for single object recognition. Due to the fovea-like attention which shifts based on task-specific objectives, the above models can be seen as having overt, top-down attention mechanisms.

An alternative approach is to alter the structure of a feed-forward network so that the convolutional activations are modified as the image moves through the network, i.e. in a bottom-up fashion. Spatial transformer networks [18] learn parameters of a transformation that can have the effect of stretching, rotating, and cropping activations between layers. Progressive Attention Networks [36] learn attention filters placed at each layer of a CNN to progressively focus on an arbitrary subset of the input, while Residual Attention Networks [41] learn feature-specific filters. Here, we consider an attentional stage that *follows* a feed-forward stage, i.e. a saliency map and image representation are produced in a feed-forward stage, then an attention mechanism determines which parts of the image representation are relevant using the saliency map.

Saliency is typically studied in the context of saliency modeling, in which a model outputs a saliency map for an image that matches human fixation data, or salient object segmentation [25]. Separately, several works have considered extracting a saliency map for understanding classification network decisions [37, 47]. Zagoruyko et al. [46] formulate a loss function that causes a student network to have similar "saliency" to a teacher network. They model saliency as a reduction operation $\mathcal{F} : \mathbb{R}^{C \times H \times W} \rightarrow \mathbb{R}^{H \times W}$ applied to a volume of convolutional activations, which we adopt due to its simplicity. Here, we investigate using a saliency map for a downstream task. Recent work has begun to explore saliency maps as inputs for prominent object detection [38] and image captioning [11], pointing to further uses of saliency-based vision models.

While we focus on using reinforcement learning for multiset classification with only class labels as annotation, RL has been applied to other computer vision tasks, including modeling eye movements based on annotated human scan paths [29], optimizing prediction performance subject to a computational budget [19], describing classification decisions with natural language [16], and object detection [28, 5, 4].

Finally, our architecture is inspired by works in hierarchical reinforcement learning. The model distinguishes between the upper level task of choosing an image region to focus on and the lower level task of classifying the object related to that region. The tasks are handled by separate networks that operate at different time-scales, with the upper level network specifying the task of the lower level network. This hierarchical modularity relates to the meta-controller / controller architecture of Kulkarni et al. [22] and feudal reinforcement learning [12, 40]. Here, we apply a hierarchical architecture to multi-label image classification, with the two levels linked by a differentiable operation.

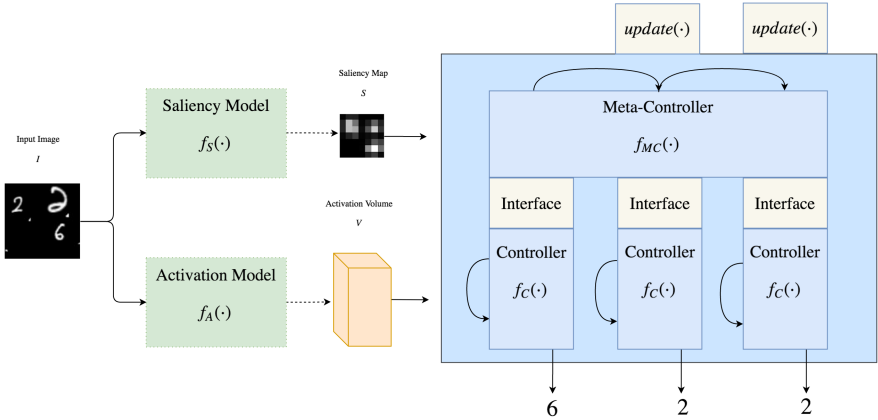

Figure 1: A high-level view of the model components. See Supplementary Materials section 3 for detailed views.

## 3 Architecture

The architecture is a hierarchical recurrent neural network consisting of two main components: the *meta-controller* and *controller*. These components assume access to a *saliency model*, which produces a saliency map from an image, and an *activation model*, which produces an activation volume from an image. Figure 1 shows the high level components, and Supplementary Materials section 3 shows detailed views of the overall architecture and individual components.

In short, given a saliency map the meta-controller places an attention mask on an object, then the controller takes subsequent glimpses and classifies that object. The saliency map is updated to account for the processed locations, and the process repeats. The meta-controller and controller operate at different time-scales; for each step of the meta-controller, the controller takes $k + 1$ steps.

**Notation** Let $\mathcal{I}$ denote the space of images, $I \in \mathbb{R}^{h_I \times w_I}$ and $\mathcal{Y} = 1, ..., n_c$ denote the set of labels. Let $\mathcal{S}$ denote the space of *saliency maps*, $S \in \mathbb{R}^{h_S \times w_S}$, let $\mathcal{V}$ denote the space of *activation volumes*, $V \in \mathbb{R}^{C \times h_V \times w_V}$, let $\mathcal{M}$ denote the space of *covert attention masks*, $M \in \mathbb{R}^{h_M \times w_M}$, let $\mathcal{P}$ denote the space of *priority maps*, $P \in \mathbb{R}^{h_M \times w_M}$, and let $\mathcal{A}$ denote an *action space*. The activation model is a function $f_A : \mathcal{I} \to \mathcal{V}$ mapping an input image to an activation volume. An example volume is the $512 \times h_V \times w_V$ activation tensor from the final conv layer of a ResNet.

**Meta-Controller** The meta-controller is a function $f_{MC} : \mathcal{S} \to \mathcal{M}$ mapping a saliency map to a covert attention mask. Here, $f_{MC}$ is a recurrent neural network defined as follows:

$$
\begin{aligned}
x_t &= [S_t, \hat{y}_{t-1}], \\
e_t &= W_{encode} x_t, \\
h_t &= \text{GRU}(e_t, h_{t-1}), \\
M_t &= \text{attn}(h_t).
\end{aligned}
$$

$x_t$ is a concatenation of the flattened saliency map and one-hot encoding of the previous step's class label prediction, and attn$(\cdot)$ is the novel spatial attention mechanism defined below. The mask is then transformed by the interface layer into a priority map that directs the controller's glimpses towards a salient region, and used to produce an initial glimpse vector for the controller.

**Gaussian Attention Mechanism** The spatial attention mechanism, inspired by covert visual attention, is a 2D discrete convolution of a mixture of Gaussians filter. Specifically, the attention mask $M$ is a $m \times n$ matrix with $M_{ij} = \phi(i, j)$, where

$$
\phi(i, j) = \sum_{k=1}^{K} \alpha^{(k)} \exp \left( -\beta^{(k)} \left[ \left( \kappa_1^{(k)} - i \right)^2 + \left( \kappa_2^{(k)} - j \right)^2 \right] \right).
$$

$K$ denotes the number of Gaussian components and $\alpha^{(k)}, \beta^{(k)}, \kappa_1^{(k)}, \kappa_2^{(k)}$ respectively denote the importance, width, and x, y center of component $k$.

To implement the mechanism, the parameters $(\alpha, \beta, \kappa_1, \kappa_2)$ are output by a network layer as a $4K$-dimensional vector $(\alpha, \beta, \kappa_1, \kappa_2)$, and the elements are transformed to their proper ranges: $\kappa_1 = \sigma(\kappa_1)m, \kappa_2 = \sigma(\kappa_2)n, \alpha = \text{softmax}(\alpha), \beta = \exp(\beta)$. Then $M$ is formed by applying $\phi$ to the coordinates $\{(i, j) \mid 1 \le i \le m, 1 \le j \le n\}$. Note that these operations are differentiable, allowing the attention mechanism to be used as a module in a network trained with back-propagation. Graves [15] proposed a 1D version; here we use a 2D version for spatial attention.

**Interface** The interface layer transforms the meta-controller's output into a priority map and glimpse vector that are used as input to the controller (diagram in Supp. Materials 3.4). The priority map combines the top-down covert attention mask with the bottom-up saliency map: $P = M \odot S$. Since $P$ influences the region that is processed next, this can also be seen as a generalization of the "winner-take-all" step in the Itti model; here a learned function chooses a *region* of high saliency rather than greedily choosing the maximum location.

To provide an initial glimpse vector $\vec{g}_0 \in \mathbb{R}^C$ for the controller, the mask is used to spatially weight the activation volume: $\vec{g}_0 = \sum_{i=1}^{h_V} \sum_{j=1}^{w_V} M_{i,j} V_{\cdot,i,j}$ This is interpreted as the meta-controller taking an initial, possibly broad and variable-sized glimpse using covert attention. The weighting produced by the attention map retains the activations around the centers of attention, while down-weighting outlying areas, effectively suppressing activations from noise outside of the attentional area. Since

the activations are averaged into a single vector, there is a trade-off between attentional area and information retention.

**Controller** The controller is a recurrent neural network $f_C : (\mathcal{P}, g_0) \to \mathcal{A}$ that runs for $k + 1$ steps and maps a priority map and initial glimpse vector from the interface layer to parameters of a distribution, and an action is sampled. The first $k$ actions select spatial indices of the activation volume, and the final action chooses a class label, i.e. $\mathcal{A}_{1,...,k} \equiv \{1, 2, ..., h_V w_V\}$ and $\mathcal{A}_{k+1} \equiv \mathcal{Y}$. Specifically:

$$
\begin{aligned}
x_i &= [P_t, \hat{y}_{t-1}, a_{i-1}, g_{i-1}], \\
e_i &= W_{encode} x_i, \\
h_i &= \mathrm{GRU}(e_i, h_{i-1}), \\
s_i &= \begin{cases} W_{location} h_i & 1 \le i \le k \\ W_{class} h_i & i = k + 1 \end{cases}, \\
p_i &= \mathrm{softmax}(s_i), \\
a_i &\sim p_i,
\end{aligned}
$$

where $t$ indexes the meta-controller time-step and $i$ indexes the controller time-step, and $a_i \in \mathcal{A}$ is an action sampled from the categorical distribution with parameter vector $p_i$. The glimpse vectors $g_i$, $i \le 1 \le k$ are formed by extracting the column from the activation volume $V$ at location $a_i = (x, y)_i$.

Intuitively, the controller uses overt attention to choose glimpse locations using the information conveyed in the priority map and initial glimpse, compiling the information in its hidden state to make a classification decision. Recall that both covert attention and priority maps are known to influence eye saccades [21]. See Supplementary Materials 3.5 for a diagram.

**Update Mechanism** During a step $t$, the meta-controller takes saliency map $S_t$ as input, focuses on a region of $S_t$ using an attention mask $M_t$, then the controller takes glimpses at locations $(x, y)_1, (x, y)_2, ..., (x, y)_k$. At step $t + 1$, the saliency map should reflect the fact that some regions have already been attended to in order to encourage attending to novel areas. While the meta-controller's hidden state can in principle prevent it from repeatedly focusing on the same regions, we explicitly update the saliency map with a function $update : \mathcal{S} \to \mathcal{S}$ that suppresses the saliency of glimpsed locations and locations with nonzero attention mask values, thereby increasing the relative saliency of the remaining unattended regions:

$$
[S_{t+1}]_{ij} = \begin{cases} 0 & \text{if } (i, j) \in \{(x, y)_1, (x, y)_2, ..., (x, y)_k\} \\ \max([S_t]_{ij} - [M_t]_{ij}, 0) & \text{otherwise} \end{cases}
$$

This mechanism is motivated by the inhibition of return effect in the human visual system; after attention has been removed from a region, there is an increased response time to stimuli in the region, which may influence visual search and encourage attending to novel areas [13, 33].

**Saliency Model** The saliency model is a function $f_S : \mathcal{I} \to \mathcal{S}$ mapping an input image to a saliency map. Here, we use a saliency model that computes a map by compressing an activation volume using a reduction operation $\mathcal{F} : \mathbb{R}^{C \times H_V \times W_V} \to \mathbb{R}^{H_V \times W_V}$ as in [46]. We choose $F(V) = \sum_{c=1}^{C} |V_i|^2$, and use the output of the activation model as $V$. Furthermore, the activation model is fine-tuned on a single-object dataset containing classes found in the multi-object dataset, so that the saliency model has high activations around classes of interest.

## 4 Learning

### 4.1 Sequential Multiset Classification

Multi-label classification tasks can be categorized based on whether the labels are lists, sets, or multisets. We claim that multiset classification most closely resembles a human's free viewing of a scene; the exact labeling order of objects may vary by individual, and multiple instances of the same object may appear in a scene and receive individual labels. Specifically, let $\mathcal{D} = \{(X_i, Y_i)\}_{i=1}^{n}$ be a dataset of images $X_i$ with labels $Y_i \subseteq \mathcal{Y}$ and consider the structure of $Y_i$.

In list-based classification, the labels $Y_i = [y_1, ..., y_{|Y_i|}]$ have a consistent order, e.g. left to right. As a sequential prediction problem, there is exactly one true label for each prediction step, so a standard

cross-entropy loss can be used at each prediction step, as in [3]. When the labels $Y_i = \{y_1, ..., y_{|Y_i|}\}$ are a set, one approach for sequential prediction is to impose an ordering $O(Y_i) \rightarrow [y_{o_1}, ..., y_{o_{|Y_i|}}]$ as a preprocessing step, transforming the set-based problem to a list-based problem. For instance, $O(\cdot)$ may order the labels based on prevalence in the training data as in [42]. Finally, *multiset* classification generalizes set-based classification to allow duplicate labels within an example, i.e. $Y_i = \{y_1^{m_1}, ...y_{|Y_i|}^{m_{|Y_i|}}\}$, where $m_j$ denotes the multiplicity of label $y_j$.

Here, we propose a training process that allows duplicate labels and is permutation-invariant with respect to the labels, removing the need for a hand-engineered ordering and supporting all three types of classification. With a saliency-based model, permutation invariance for labels is especially crucial, since the most salient (and hence first classified) object may not correspond to the first label.

## 4.2 Training

Our solution is to frame the problem in terms of maximizing a non-smooth reward function that encourages the desired classification and attention behavior, and use reinforcement learning to maximize the expected reward. Assuming access to a trained saliency model and activation model, the meta-controller and controller can be jointly trained end-to-end.

**Reward** To support multiset classification, we propose a multiset-based reward for the controller's classification action. Specifically, consider an image $X$ with $m$ labels $\mathcal{Y} = \{y_1, ..., y_m\}$. At meta-controller step $t$, $1 \leq t \leq m$, let $\mathcal{A}_i$ be a multiset of available labels, let $f_i(X)$ be the corresponding class scores output by the controller. Then define:

$$R_{i_{\text{clf}}} = \begin{cases} +1 & \text{if } \hat{y}_i \in \mathcal{A}_i \\ -1 & \text{otherwise} \end{cases} \qquad \mathcal{A}_{i+1} = \begin{cases} \mathcal{A}_i \setminus \hat{y}_i & \text{if } \hat{y}_i \in \mathcal{A}_i \\ \mathcal{A}_i & \text{otherwise} \end{cases}$$

where $\hat{y}_i \sim \text{softmax}(f_i(X))$ and $\mathcal{A}_1 \leftarrow \mathcal{Y}$. In short, a class label is sampled from the controller, and the controller receives a positive reward if and only if that label is in the multiset of available labels. If so, the label is removed from the available labels. Clearly, the reward for sampled labels $\hat{y}_1, \hat{y}_2, .., \hat{y}_m$ equals the reward for $\sigma(\hat{y}_1), \sigma(\hat{y}_2), .., \sigma(\hat{y}_m)$ for any permutation $\sigma$ of the $m$ elements. Note that list-based tasks can be supported by setting $\mathcal{A}_i \leftarrow y_i$.

The controller's location-choice actions simply receive a reward equal to the priority map value at the glimpse location, which encourages the controller to choose locations according to the priority map. That is, for locations $(x, y)_1, ..., (x, y)_k$ sampled from the controller, define $R_{i_{\text{loc}}} = P_{(x,y)_i}$.

**Objective** Let $n = 1...N$ index the example, $t = 1...M$ index the meta-controller step, and $i = 0...K$ index the controller step. The goal is choosing $\theta$ to maximize the total expected reward: $J(\theta) = \mathbb{E}_{p(\tau|f_\theta)} \left[ \sum_{n,t,i} R_{n,t,i} \right]$ where the rewards $R_{n,t,i}$ are defined as above, and the expectation is over the distribution of trajectories produced using a model $f$ parameterized by $\theta$. An unbiased gradient estimator for $\theta$ can be obtained using the REINFORCE [43] estimator within the stochastic computation graph framework of Schulman et al. [34] as follows.

Viewed as a stochastic computation graph, an input saliency map $S_{n,t}$ passes through a path of deterministic nodes, reaching the controller. Each of the controller's $k+1$ steps produces a categorical parameter vector $p_{n,t,i}$ and a stochastic node is introduced by each sampling operation $a_{t,i} \sim p_{n,t,i}$. Then form a surrogate loss function $\mathcal{L}(\theta) = \sum_{t,i} \log p_{t,i} R_{t,i}$ with the stochastic computation graph. By Corollary 1 of [34], the gradient of $\mathcal{L}(\theta)$ gives an unbiased gradient estimator of the objective, which can be approximated using Monte-Carlo sampling: $\frac{\partial}{\partial \theta} J(\theta) = \mathbb{E} \left[ \frac{\partial}{\partial \theta} \mathcal{L}(\theta) \right] \approx \frac{1}{B} \sum_{b=1}^{B} \frac{\partial}{\partial \theta} \mathcal{L}(\theta)$. As is standard in reinforcement learning, a state-value function $V(s_{t,i})$ is used as a baseline to reduce the variance of the REINFORCE estimator, thus $\mathcal{L}(\theta) = \sum_{t,i} \log p_{t,i}(V(s_{t,i}) - R_{t,i})$. In our implementation, the controller outputs the state-value estimate, so that $s_{t,i}$ is the controller's hidden state.

# 5 Experiments

We validate the classification performance, training process, and hierarchical attention with set-based and multiset-based classification experiments. To test the effectiveness of the permutation-invariant

Table 1: Metrics on the test set for MNIST Set and Multiset tasks, and SVHN Multiset.

| | MNIST Set | | MNIST Multiset | | SVHN Multiset | |
| --- | --- | --- | --- | --- | --- | --- |
| | F1 | 0-1 | F1 | 0-1 | F1 | 0-1 |
| HSAL-RL | **0.990** | **0.960** | **0.978** | **0.935** | **0.973** | **0.947** |
| Cross-Entropy | 0.735 | 0.478 | 0.726 | 0.477 | 0.589 | 0.307 |

RL training, we compare against a baseline model that uses a cross-entropy loss on the probabilities $p_{t,i}$ and (randomly ordered) labels $y_i$ instead of the RL training, similar to training proposed in [42].

**Datasets** Two synthetic datasets, MNIST Set and MNIST Multiset, as well as the real-world SVHN dataset, are used. For MNIST Set and Multiset, each 100x100 image in the dataset has a variable number (1-4) of digits, of varying sizes (20-50px) and positions, along with cluttering objects that introduce noise. Each label in an image from MNIST Set is unique, while MNIST Multiset images may contain duplicate labels. Each dataset is split into 60,000 training examples and 10,000 testing examples, and metrics are reported for the testing set. SVHN Multiset consists of SVHN examples with label order randomized when a batch is sampled. This removes the natural left-to-right order of the SVHN labels, thus turning the classification into a multiset task.

**Evaluation Metrics** To evaluate classification performance, macro-F1 and exact match (0-1) as defined in [26] are used. For evaluating the hierarchical attention mechanism we use visualization as well as a saliency metric for the controller's glimpses, defined as $\text{attn}_{\text{saliency}} = \frac{1}{k} \sum_{i=1}^{k} S_{ti}$ for a controller trajectory $(x, y)_1, ..., (x, y)_k, \hat{y}_t$ at meta-controller time step $t$, then averaged over all time steps and examples. A high score means that the controller tends to pick salient points as glimpse locations.

**Implementation Details** The activation and saliency model is a ResNet-34 network pre-trained on ImageNet. For MNIST experiments, the ResNet is fine-tuned on a single object MNIST Set dataset, and for SVHN is fine-tuned by randomly selecting one of an image's labels each time a batch is sampled. Images are resized to 224x224, and the final (4th) convolutional layer is used ($V \in \mathbb{R}^{512 \times 7 \times 7}$). Since the label sets vary in size, the model is trained with an extra "stop" class, and during inference greedy argmax sampling is used until the "stop" class is predicted. See Supplementary Materials section 1 for further details.

## 5.1 Experimental Evaluation

In this section we analyze the model's classification performance, the contribution of the proposed RL training, and the behavior of the hierarchical attention mechanism.

**Classification Performance** Table 1 shows the evaluation metrics on the set-based and multiset-based classification tasks for the proposed hierarchical saliency-based model with RL training ("HSAL-RL") and the cross-entropy baseline ("Cross-Entropy") introduced above. HSAL-RL performs well across all metrics; on both the set and multiset tasks the model achieves very high precision, recall, and macro-F1 scores, but as expected, the multiset task is more difficult. We conclude that the proposed model and training process is effective for these set and multiset image classification tasks.

**Contribution of RL training** As seen in Table 1, performance is greatly reduced when the standard cross-entropy training is used, which is not invariant to the label ordering. This shows the importance of the RL training, which only assumes that predictions are *some* permutation of the labels.

**Controller Attention** Based on $\text{attn}_{saliency}$, the controller learns to glimpse in salient regions more often as training progresses, starting at 58.7 and ending at 126.5 (see graph in Supplementary Materials Section 2). The baseline, which does not have the reward signal for its glimpses, fails to improve over training (remaining near 25), demonstrating the importance and effectiveness of the controller's glimpse rewards.

**Hierarchical Attention Visualization** Figure 2 visualizes the hierarchical attention mechanism on three example inference processes. See Supplementary Materials Section 4 for more examples, which we discuss here. In general, the upper level attention highlights a region encompassing a digit, and

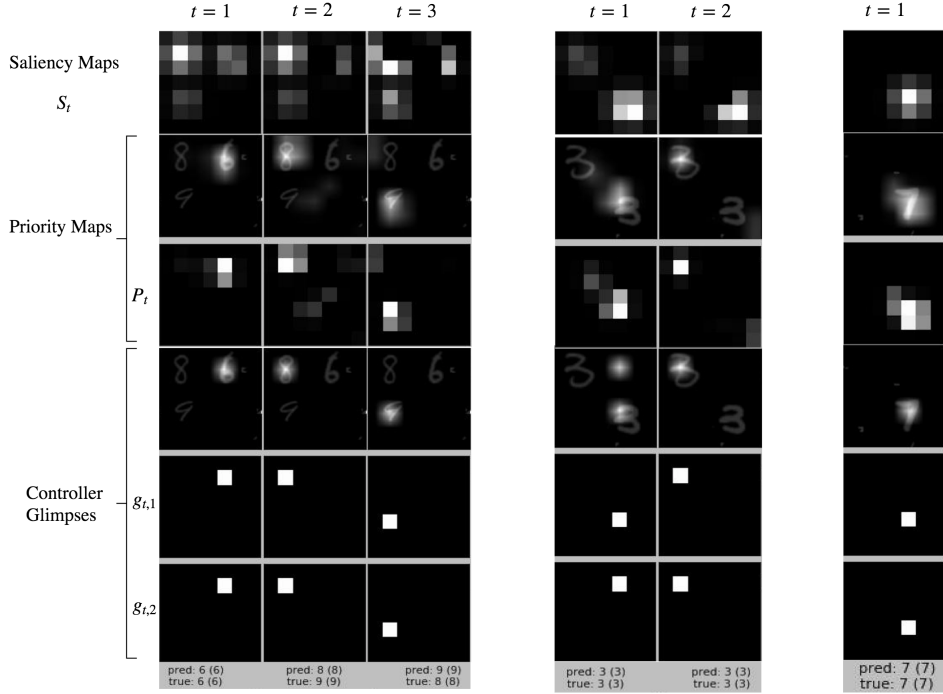

Figure 2: The inference process showing the hierarchical attention on three different examples. Each column represents a single meta-controller step, two controller glimpses, and classification.

the lower level glimpses near the digit before classifying. Notice the saliency map update over time, the priority map's structure due to the Gaussian attention mechanism, and the variable-sized focus of the priority map followed by finer-grained glimpses. Note that the predicted labels need not be in the same order as the ground truth labels (e.g. "689"), and that the model can predict multiple instances of a label (e.g. "33", "449"), illustrating multiset prediction. In some cases, the upper level attention is sufficient to classify the object without the controller taking related glimpses, as in "373", where the glimpses are in a blank region for the 7. In "722", the covert attention is initially placed on both the 7 and the 2, then the controller focuses only on the 7; this can be interpreted as using the multiple spotlight capability of covert attention, then directing overt attention to a single target.

## 5.2 Limitations

**Saliency Map Input** Since the saliency map is the only top-level input, the quality of the saliency model is a potential performance bottleneck. As Figure 4 shows, in general there is no guarantee that all objects of interest will have high saliency relative to the locations around them. However, the modular architecture allows for plugging in alternative, rigorously evaluated saliency models such as a state-of-the-art saliency model trained with human fixation data [10].

**Activation Resolution** Currently, the activation model returns the highest-level convolutional activations, which have a 7x7 spatial dimension for a 224x224 image. Consider the case shown in Figure 3. Even if the controller acted optimally, activations for multiple digits would be included in its glimpse vector due to the low resolution. This suggests activations with higher spatial resolution are needed, perhaps by incorporating dilated convolutions [45] or using lower-level activations at attended areas, motivated by covert attention's known enhancement of spatial resolution [6, 7, 14].

## 6 Conclusion

We proposed a novel architecture, attention mechanism, and RL-based training process for sequential image attention, supporting multiset classification. The proposal is a first step towards incorporating notions of saliency, covert and overt attention, and sequential processing motivated by the biological visual attention literature into deep learning architectures for downstream vision tasks.

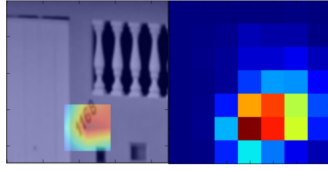

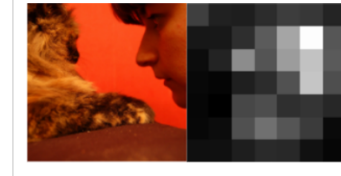

Figure 3: The location of highest saliency from a 7x7 saliency map (right) is projected onto the 224x224 image (left).

Figure 4: The cat is a label in the ground truth set but does not have high salience relative to its surroundings.

**Acknowledgments**

This work was partly supported by the NYU Global Seed Funding <Model-Free Object Tracking with Recurrent Neural Networks>, STCSM 17JC1404100/1, and Huawei HIPP Open 2017.

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
