[Supplementary Material · supplementary-materials-saliency.pdf]

# Supplementary Materials: Saliency-based Sequential Image Attention with Multiset Prediction

**Sean Welleck**
New York University
wellecks@nyu.edu

**Jialin Mao**
New York University
jialin.mao@nyu.edu

**Kyunghyun Cho**
New York University
kyunghyun.cho@nyu.edu

**Zheng Zhang**
New York University
zz@nyu.edu

## 1    Additional Implementation Details

The meta-controller and controller encode their input to 128 and 512-dimensional vectors, respectively, and use a GRU with hidden sizes 128 and 512, respectively. The Gaussian attention uses 5 components, with each $\beta$ constrained to $[1.5, 2.0]$. The priority maps and the input saliency map are normalized to have maximum 1 and minimum 0. The number of glimpses $k$ is fixed to 2. The classification scores and $V(s_{t,i})$ are linear transformations of the GRU hidden state. Action scores are computed from the GRU hidden state with a 1 layer MLP with hidden dimension 64. $V(s_{t,i})$ is trained to minimize MSE with the observed reward. The gradients of the action score MLP and state-value linear transformation are detached from the rest of the model during back-propagation. To accelerate training, a supervised cross-entropy loss was added to HSAL-RL for stop-class prediction. That is, for a label set of cardinality $m$, on the $m + 1$st meta-controller step a cross-entropy loss is attached to the controller's softmax probabilities, using the "stop" label as the ground-truth label for the loss.

## 2    Training Curves

Figure 1: attn$_{\text{saliency}}$ for the controller's glimpses over the course of training for HSAL-RL (green) and the baseline (teal) on the MNIST Multiset task.

Figure 2: Macro-F1 for HSAL-RL (green) and the baseline (teal) on the MNIST Multiset task over the course of training. OP, OR, 0-1 metrics have an analogous shape.

## 3    Architecture Diagrams

Below are detailed diagrams for the full architecture and individual components.

## 3.1 Full Architecture

Figure 3: The full architecture, expanded for one meta-controller time-step. As input to the system, an initial saliency map and activation volume are generated by the saliency model and activation models, respectively. At each meta-controller step, an updated saliency map is mapped to a covert attention mask. The interface forms an initial glimpse vector and priority map using the attention mask. Based on the priority map and initial glimpse vector, the controller chooses $k$ glimpse locations, then classifies. Note that $k + 1$ controller steps occur for every meta-controller step.

## 3.2 Meta-Controller

Figure 4: An implementation of the meta-controller.

## 3.3 Update

Figure 5: A conceptual diagram of the $update(\cdot)$ function.

## 3.4 Interface

Figure 6: An implementation of the interface layer.

## 3.5 Controller

Figure 7: An implementation of the controller.

# 4  Attention Visualizations

Figure 8: Additional hierarchical attention visualizations, discussed in the Experimental Evaluation.