[Reviews · NeurIPS 2017]

Reviewer 1



In this paper, the authors proposed a hierarchical visual architecture that operates on a saliency map and uses a novel attention mechanism based on 2D Gaussian model. Furthermore this mechanism sequentially focuses on salient regions and takes additional glimpses within those regions in multi-label image classification. This sequential attention model also supports multiset prediction, where a reinforcement learning based training procedure allows classification to be done on instances with arbitrary label permutation and multiple instances per label. Pros: 1) This paper proposes a novel saliency based attention mechanism that utilizes saliency in the top layer (meta-controller) with a new 2D Gaussian based attention map. This new attention map models the regional /positional 2D information with a mixture of Gaussian distributions, which is more general than the standard attention layer (in DRAW, Show-attend-tell), where attention is enforced based on softmax activation. 2) The interface layer fuses the attentional inference and the hidden state to produce a priority map (which is a 2D map that is filtered by attention), and a glimpse vector for estimating glimpse location for the controller. This mechanism is intuitive as it's inspired by human-level attention mechanism. 3) The controller section combines the priority map and the glimpse vector, and it performs a k+1 step estimation, where the first k step refines the glimpse location (in terms of the glimpse vector), and the last step predicts the classification label. This architecture further refines the attention capability, and it improves the accuracy of the multi-label classification. 4) A multi-object (multi-set) multi-label classification problem is studied by the proposed architecture using a reinforcement learning based training process. This allows classification with arbitrary label permutation (changes sequentially I presume) and multiple instances per label. The simple experiments on MINST also demonstrate the effectiveness of sequential multi-set prediction with this attention mechanism. Cons: 1) The reference section is poorly written, please tidy it up in the final version. 2) The experiment of multi-set classification is interesting. But to the best of my knowledge, such a problem setting (the set-based prediction discussed in section 4.1) is sequential by nature (in the ordering part). May be I am missing something here, but since it is specific-tuned for demonstrating the effectiveness of sequential prediction, I am not sure how the comparison with binary-crossetnropy is justified here. Also, how does this approach compare with RAM, another popular attention based algorithm that has a sequential decision-making/RL flavor? On the overall, I found this paper well-written and intuitive, as the hierarchical attention mechanism is inspired by how humans process visual scenes selectively and sequentially with attention. I think this paper has significant contribution to the work of attention and I would recommend its acceptance as a NIPS paper.

Reviewer 2



This submission proposes a recurrent attention architecture (composed of multiple steps) that operates on a saliency map that is computed with a feed forward network. The proposed architecture is based on the biological model of covert attention and overt attention. This is translated into a meta-controller (covert attention) that decides which regions need overt attention (which are then treated by lower-level controllers that glimpse in the chosen regions). This is implemented as a form of hierarchical reinforcement learning, where the reward function for the meta-controller is explicitly based on the idea of multiset labels. The (lower-level) controllers receive reward for directing attention to the area with high priority, as output by a previously computed priority map. The whole architecture is trained using REINFORCE, a standard RL algorithm. The proposed architecture is used for multi-label, specifically multi-set setting where each image can have multiple labels, and labels may appear multiple times. The experiments are conducted on MNIST and MNIST multiset datasets, where images contain one single digit or multiple digits. The paper is well written and the model is well explained. The detailed visualizations in the supplementary material are required to understand the architecture with many components. The details of the architecture when considered with the supplementary material, are well motivated and explained; the paper is comprehensive enough to allow the reproduction of results. The topic is interesting and relevant for the community. Some questions are as follows: - The “meta controller” unit takes a saliency map and the class label prediction from the previous step as inputs and learns a covert attention mask. How is the saliency map initialized in the first step? Which saliency map does the “interface” unit use, the saliency map learned in the previous step or the initial one? - The controller takes the priority map and the glimpse vector as inputs, runs them through a recurrent net (GRU) and each GRU unit outputs a label. How are the number of GRU units determined here? Using the number of gaussians learned by the gaussian attention mechanism? Is this the same as the number of peaks in the priority map? A missing related work would be Hendricks et.al., Generating Visual Explanations, ECCV 2016 that also trains a recurrent net (LSTM) using REINFORCE algorithm to generate textual explanations of the visual classification decision. However it does not contain an integrated visual attention mechanism. The main weakness of the paper is the experimental evaluation. The only experimental results are reported on synthetic datasets, i.e. MNIST and MNIST multi-set. As the objects are quite salient on the black background, it is difficult to judge the advantage of the proposed attention mechanism. In natural images, as discussed in the limitations, detecting saliency with high confidence is an issue. However, this work having been motivated partially as a framework for an improvement to existing saliency models should have been evaluated in more realistic scenarios. Furthermore, the proposed attention model is not compared with any existing attention models. Also, a comparison with human gaze based as attention (as discussed in the introduction) would be interesting. A candidate dataset is CUB annotated with human gaze data in Karessli et.al, Gaze Embeddings for Zero-Shot Image Classification, CVPR17 (another un-cited related work) which showed that human gaze based visual attention is class-specific. Minor comment: - The references list contains duplicates and the publication venues and/or the publication years of many of the papers are missing.